# Commercial Potential of the Cyanobacterium *Arthrospira maxima*: Physiological and Biochemical Traits and the Purification of Phycocyanin

**DOI:** 10.3390/biology11050628

**Published:** 2022-04-20

**Authors:** Jihae Park, Hojun Lee, Thai Binh Dinh, Soyeon Choi, Jonas De Saeger, Stephen Depuydt, Murray T. Brown, Taejun Han

**Affiliations:** 1Development & Planning Office, Ghent University Global Campus, 119-5, Songdomunhwa-ro, Incheon 21985, Korea; jihae.park@ghent.ac.kr (J.P.); hojun.lee@ugent.be (H.L.); 2Laboratory of Plant Growth Analysis, Ghent University Global Campus, 119-5, Songdomunhwa-ro, Incheon 21985, Korea; stephen.depuydt@ghent.ac.kr; 3Department of Cosmetic Science and Management, Incheon National University, 119, Academy-ro, Incheon 22012, Korea; thaibinhnt2@gmail.com; 4Department of Marine Science, Incheon National University, 119, Academy-ro, Incheon 22012, Korea; chlthdus0501@hanmail.net; 5Department of Plant Biotechnology and Bioinformatics, Ghent University, Technologiepark 71, 9052 Ghent, Belgium; jonas.desaeger@ghent.ac.kr; 6School of Marine Science & Engineering, Plymouth University, Plymouth PL4 8AA, Devon, UK; m.t.brown@plymouth.ac.uk; 7Department of Animal Sciences and Aquatic Ecology, Ghent University, Coupure Links 653-Block F, 9000 Ghent, Belgium

**Keywords:** pH, temperature, *Arthrospira maxima*, photosynthetic performance, phycocyanin

## Abstract

**Simple Summary:**

*Arthrospira maxima* is an unbranched, filamentous cyanobacterium rich in important cellular products such as vitamins, minerals, iron, essential amino acids, essential fatty acids, and protein, which has made it one of the most important commercial photoautotrophs. To optimize the growth conditions for the production of target compounds and to ensure profitability in commercial applications, the effects of pH and temperature were investigated. *A. maxima* has been shown to be tolerant to a range of pH conditions and to exhibit hyper-accumulation of phycoerythrin and allophycocyanin at low temperatures. These traits may offer significant advantages for future exploitation, especially in outdoor cultivation with fluctuating pH and temperature. Our study also demonstrated a new method for the purification of phycocyanin from *A. maxima* by using by ultrafiltration, ion-exchange chromatography, and gel filtration, producing PC at 1.0 mg·mL^−1^ with 97.6% purity.

**Abstract:**

*Arthrospira maxima* is a natural source of fine chemicals for multiple biotechnological applications. We determined the optimal environmental conditions for *A. maxima* by measuring its relative growth rate (RGR), pigment yield, and photosynthetic performance under different pH and temperature conditions. RGR was highest at pH 7–9 and 30 °C. Chlorophyll *a*, phycocyanin, maximal quantum yield (*F_v_*/*F_m_*), relative maximal electron transport rate (rETR_max_), and effective quantum yield (Φ_PSII_) were highest at pH 7–8 and 25 °C. Interestingly, phycoerythrin and allophycocyanin content was highest at 15 °C, which may be the lowest optimum temperature reported for phycobiliprotein production in the *Arthrospira* species. A threestep purification of phycocyanin (PC) by ultrafiltration, ion-exchange chromatography, and gel filtration resulted in a 97.6% purity of PC.

## 1. Introduction

Cyanobacteria in the genus *Arthrospira* (formerly Spirulina) are easy to culture and harvest, exhibit high growth rates, and produce a variety of compounds used in natural food supplements, therapeutics, diagnostics, and biofuels [1,2,3,4]. *Arthrospira maxima* Setchell and Gardner is an unbranched, filamentous cyanobacterium rich in important cellular products such as vitamins, minerals, iron, essential amino acids, essential fatty acids, and protein [5]. The growing international demand for high-value phytonutrients and pigments produced by *A. maxima* has made it one of the most important commercial photoautotrophs, together with *Chlorella* and *A. platensis* [6,7]. 

The development of a biotechnological marine product for commercial use involves a series of steps that require (i) a large amount of biomass, (ii) high concentrations of the desired product in the biomass, and (iii) an industrial-scale process [8]. Therefore, to realize the full potential, it is essential to optimize the growth conditions and production of the target compound(s), which could be achieved through empirical studies. The biochemical composition of cyanobacteria varies under different culture conditions, and optimizing growth conditions for the production of target compounds is an important step for ensuring profitability in commercial applications [9]. Among the various environmental factors that influence processes such as growth, development, reproduction, and photosynthesis in autotrophs, the quantity (irradiance) and quality (spectral composition) of light are essential. Light also affects the production of metabolites [10]. In particular, pH and temperature are the most critical environmental factors influencing algal biotechnological processes [11]. pH alters nutrient solubility and bioavailability, membrane transport, enzyme activity, electron transport, and osmotic balance in the cytosol [12,13]. Cyanobacteria require highly alkaline conditions for optimal growth, and negative effects of low pH have been observed experimentally among different taxa [11]. Understanding the effects of pH on cyanobacteria has gained importance owing to the acidification of aquatic systems driven by climate change; this could negatively impact natural populations of cyanobacteria unless they exhibit protective or adaptive mechanisms [14]. Temperature also affects many cyanobacterial processes, including metabolism, growth, photosynthesis, reproduction, distribution, and survival. Many cellular processes of cyanobacteria are temperature-dependent, with optima between 25 °C and 40 °C [15]. Low temperatures negatively affect metabolism, whereas high temperatures inhibit growth [16]. Climate change resulting from global warming may be advantageous for cyanobacteria, as most cyanobacterial species show optimal growth at temperatures of 25 °C [17,18] or higher [19]. 

Phycobiliproteins are important pigments constituting approximately 40% of the soluble proteins in cyanobacteria [20]. Three types of phycobiliproteins are produced by cyanobacteria: bluish green allophycocyanin (APC), blue phycocyanin (PC), and red phycoerythrin (PE). Phycobiliproteins are water soluble and fluorescent, making them useful as natural coloring agents in foods, health drinks, and cosmetics [21,22] and as fluorescent labels in cellular assays [23]. They were also recently found to have important applications in human health owing to their anti-inflammatory, antioxidant, and anticancer properties [24]. The price of phycobiliproteins varies from USD 3 (per 25 mg of food/cosmetic grade pigments) to USD 1500 (per 1 mg of highly purified molecular markers labelled with antibodies or other fluorescent molecules) [25]. In 2018, the value of these pigments in the commercial sector was estimated to reach USD 1830 million by 2023 [26]. Even though *A. maxima* has a higher protein content than *Arthrospira platensis* [27], there have been few studies on the commercial production of phycobiliproteins from *A. maxima* [28]. 

PC is the major constituent of phycobilisomes in cyanobacteria and is water soluble and highly fluorescent. Its outstanding light absorbance and reddish fluorescence make PC one of the most important value-added products of cyanobacteria, being widely used for tracers in fluorescence immunoassays and in microscopy for diagnostics and biomedical research [29,30]. 

In this study, we examined the effects of pH and temperature on growth rate, pigment composition, and Chl *a* fluorescence of *A. maxima*. Our results will help to determine the optimal conditions for the production of *A. maxima* bioproducts and to predict cyanobacterial responses to acidification and warming driven by climate change. We also tested a three-step ultrafiltration, ion exchange, and gel filtration method for improving the purification of phycobiliproteins (especially PC).

## 2. Materials and Methods

### 2.1. A. maxima Culture

*A. maxima* strains (LIMS-PS-1691) were obtained from the Korea Marine Microalgae Culture Center and cultured in glass flasks (500 mL), each containing 400 mL of SAG medium (Austin, TX, USA). Cultures were maintained at 25 °C, pH 8.5, and a salinity of 20 psu. Cultures were exposed to 60 μmol m^−2^·s^−1^ photon flux density (PFD) light, with a 14/10 h light/dark cycle. Square LED (light emitting diode) panel lights (340 × 500 × 10 mm; Daewo, Bucheon, Korea) were used as main source of illumination for the growth of *A. maxima*. Each LED strip comprised 20 diodes spaced at 1 cm intervals vertically and horizontally. PFDs were measured with a quantum sensor (Licor-1400, Li-Cor, Lincoln, NE, USA).

### 2.2. Treatments

Treatments consisted of eight different pH levels (pH 4, 5, 6, 7, 8, 9, 10, and 11 (±0.1)) and seven temperatures (5 °C, 10 °C, 15 °C, 20 °C, 25 °C, 30 °C, and 35 °C (±1 °C)). Flasks containing 200 mL of *A. maxima* culture were exposed to each treatment. pH was adjusted using 1 M NaOH or 1 M HCl. All flasks were kept in environmentally controlled growth chambers. While testing the effect of one environmental factor, all other conditions, including pH, temperature, salinity, PFD, and photoperiod, were kept constant as described above. No time–series of pH monitoring was carried out.

### 2.3. Assessment

#### 2.3.1. Growth Analysis

Each 200 mL culture was harvested after 8 d. Cultures were centrifuged at 5974× *g* for 10 min, followed by washing with distilled water and another round of centrifugation. Pellets were then dried at 60 °C until their weight remained constant. The relative growth rate (RGR) of *A. maxima* was calculated as follows:$$\mathrm{RGR}\left({\%\;\mathrm{day}}^{-1}\right)=\frac{\left(lnDW_{f}-lnDW_{i}\right)}{t_{f}}\times 100$$
where *DW_i_* represents the initial dry weight biomass (288 ± 7 mg·L^−1^), *DW_f_* represents the final dry weight biomass, and *t_f_* indicates the test duration. RGR is an instantaneous rate estimate, and thus simplifies much of the variability in growth over time. Nevertheless, RGR is a useful metric for the relative productivity of species growing under given conditions. Photoperiods for all samples were synchronized, thus preventing the harvesting of samples at different stages in their circadian rhythms.

#### 2.3.2. Pigment Analysis

For each treatment, one of the 200 mL cell cultures was harvested after 8 d, washed with distilled water as above, and resuspended in 90% acetone. Acetone extracts were incubated for 30 min at 4 °C in the dark and then centrifuged at 5974× *g* for 5 min. The acetone supernatant was collected, and the extraction procedure was repeated five times until clear acetone extracts were obtained. Absorption of supernatants was measured at 664 nm. Chl *a* concentration was calculated as follows [31]:$$\mathrm{Chl}\;a\;\left(\mathrm{mg}\cdot g^{-1}\right)=A_{664}/73.6$$

To measure the concentrations of phycobiliproteins, one of the 200 mL cell cultures from each treatment was harvested after 8 d, centrifuged at 5974× *g* for 5 min, rinsed twice with distilled water, and then resuspended in 2 mL of sodium phosphate buffer (0.1 M, pH 6.8). Extracts were subsequently centrifuged at 5974× *g* for 5 min at 4 °C and clear supernatants containing phycobiliproteins were collected. The extraction process was repeated five times. The absorption of supernatants was measured at wavelengths ranging from 250 to 700 nm using a UV-Vis spectrophotometer (Scinco S-3100 PDA, Seoul, Korea). Concentrations of phycobiliproteins were calculated using the following equations [32]:$$\mathrm{PC}\;\left(\mathrm{mg}\cdot g^{-1}\right)=\left(A_{620}-0.474\times A_{652}\right)/5.34$$
$$\mathrm{APC}\;\left(\mathrm{mg}\cdot g^{-1}\right)=\left(A_{620}-0.208\times A_{620}\right)/5.09$$
$$\mathrm{PE}\;\left(\mathrm{mg}\cdot g^{-1}\right)=\left[A_{562}-2.41\left(PC\right)-0.849\left(APC\right)\right]/9.62$$

#### 2.3.3. Photosynthesis Analysis

Chl *a* fluorescence was measured as a proxy for photosynthesis using Imaging PAM (Walz, Germany). Algal samples were first incubated in the dark at 25 ± 1 °C for 10–15 min. Samples were subsequently exposed to pulses of approximately 0.15 µmol photon m^−2^·s^−1^ from a light emitting diode (LED), and the initial fluorescence yield (*F_o_*) was recorded. Samples were then exposed to a 5000 µmol photons m^−2^·s^−1^ saturation pulse to measure the maximum fluorescence yield (*F_m_*). *F_o_* estimates baseline fluorescence when all photosystem II (PSII) reaction centers are open, while *F_m_* represents fluorescence when all reaction centers are closed. *F_v_*/*F_m_* was calculated as follows:$$F_{v}/F_{m}=\left(F_{m}-F_{0}\right)/F_{m}$$
where *F_v_* is the variable fluorescence.

Rapid light curves were developed using 10 s pulses of actinic light, which increased in intensity from 0 to 1517 μmol photons m^−2^·s^−1^ [33]. The effective quantum yield (Φ_PSII_) was calculated as follows:$$\Phi_{\mathrm{PSII}}=\left({F^{\prime }}_{m}-F\right)/F^{\prime }m$$
where *F*′*_m_* is the maximum light-acclimated fluorescence and *F* is the fluorescence at a given light intensity.

The relative electron transport rate (rETR) was calculated as the slope described by each Φ_PSII_ multiplied by its PFD plotted against PFD. The ETR is given in arbitrary units since the absorbance of the microalgal sample was not measured. The relative maximum electron transport rate (rETR_max_) was calculated using the following equation adapted from Jassby and Platt [34]:$$\mathrm{rETR}={\mathrm{ETR}}_{\max}\times \tan h\left(\alpha \times I/{\mathrm{ETR}}_{\max}\right)$$
where *α* indicates the ETR under limited light and *I* indicates the PFD under limited light.

### 2.4. Purification of PC

#### 2.4.1. Fractionation by Ultrafiltration

After 4 weeks under the culture conditions described above, *A. maxima* was harvested and centrifuged at 15,000× *g* rpm for 15 min at 4 °C. The pelleted cells (8 mL, 613 mg) were collected and mixed with 40 mL of sodium phosphate solution (pH 6.0). The mixed solution was incubated for 24 h at 4 °C with gentle stirring. The supernatant was then collected and filtered through Whatman filter paper (No. 541, 110 mm) to remove unpurified components. To prevent protein denaturation and ensure a stable sample for the next purification process, a buffer exchange system was operated with sodium phosphate buffer (pH 7.0) using a dialysis membrane (MWCO: 12–14,000) for 24 h. To prepare highly concentrated PC protein from the sample subjected to buffer exchange, the supernatant and filtrate were separated using an ultra-centrifugal filter with a molecular weight cut-off of 10,000 (Fisher Scientific, Hampton, NH, USA), and the concentrated supernatant was obtained.

#### 2.4.2. Separation by Ion Exchange

Ion exchange was used to separate PC proteins according to the charge properties of PC and other proteins. After fractionation and concentration, 3.1 mg·mL^−1^ of concentrated supernatant was collected and adsorbed onto 70 mL of DEAE-Sepharose beads by stirring for 24 h at 4 °C to increase adsorption strength. The DEAE-Sepharose beads adsorbed with PC protein were then introduced into a DEAE column (DFF100, Sigma-Aldrich, St. Louis, MO, USA) by gravity. After removal of the unbound proteins using washing buffer, the PC protein bound to the DEAE-Sepharose beads was eluted using elution buffer spiked with 0.1 M NaCl; each eluate was collected at a flow rate of 20 mL·h^−1^, and PC protein was identified by absorbance at 620 nm using an ELISA reader.

#### 2.4.3. Further Separation by Gel Filtration

The sample separated by the second step ion exchange (concentration 2.30 mg·mL^−1^) was loaded onto a column packed with Sephacryl S-200 beads. Since separation by gel filtration is not affected by the buffer, PC protein passing through the Sephacryl S-200 column was eluted using the same sodium phosphate buffer (pH 7.0) as in step 1. Absorbance of samples aliquoted for each section was measured at 620 nm, while PC protein was detected using an ELISA reader.

The purity of the PC protein obtained using the three step purification process was confirmed by automated analysis using TOTALLAB software (TotalLab Ltd., Newcastle upon Tyne, UK). An objective analysis of the purified proteins obtained from the three step separation process was performed by the Korea Institute of Basic Science (KBSI, https://www.kbsi.re.kr/eng, accessed on 6 April 2022; issue number: MS-00020-1) using HPLC (Agilent 1200 Series with Aeries WIDEPORE (3.6 µm) C4 column (150 × 4.6 mm)). The solvents used were 0.1% trifluoroacetic acid in water and 0.075% trifluoroacetic acid in acetonitrile, and the flow rate was 1 mL·min^−1^ at a 40 °C column temperature.

#### 2.4.4. Sodium Dodecyl Sulfate–Polyacrylamide Gel Electrophoresis (SDS–PAGE)

Fractions of PC extracts from the different extraction and purification steps were characterized using SDS-PAGE to determine the molecular mass of PC. SDS–PAGE was performed in a vertical chamber using 12.5% polyacrylamide gel containing 0.1% SDS, 4% acrylamide, and 0.1% bisacrylamide. Electrophoresis was performed at room temperature and the proteins visualized using Coomassie brilliant blue staining. The molecular mass of the protein band was determined using a calibration gel with standard proteins as the molecular mass markers (Sigma SDS).

### 2.5. Statistical Analysis

One-way analysis of variance (ANOVA) was performed for treatments on each response variable after checking for homogeneity. There were six replicates per treatment per measure. Post hoc comparisons were performed using the least significant difference (LSD) test and a significance level of *p* < 0.05.

## 3. Results and Discussion

### 3.1. Growth

pH is an important regulator of cellular processes, including the solubility and bioavailability of CO_2_ and nutrients, membrane permeability, enzyme activity, photosynthesis, and osmoregulation of the cytoplasm [13,35]. The RGR of *A. maxima* showed a clear optimum at intermediate pH (Figure 1A). The growth rate of *A. maxima* was lowest (7.83%·d^−1^) at pH 4 and highest (mean growth rate = 12.1–13.1%·d^−1^) at pH 7–9. At pH 5, 6, 10, and 11, the growth rates of *A. maxima* were similar, with mean values ranging from 9.8 to 11.2%·d^−1^.

Cyanobacteria are alkalophiles with typical pH optimums ranging from pH 6 to 10, with more extreme values promoting leaching [11,36,37,38]. Rafiqul, Jalal, and Alam [11] reported that pH levels of 9.0 and 10.0 were optimal for the growth of *A. platensis* and *Arthrospira fusiformis*, respectively. Thirumala [38] reported the optimal growth of *A. platensis* at pH 10.0. For *A. maxima*, optimal growth was previously reported between pH 6 and 10 [6]. In the present study, we observed the highest growth rate of *A. maxima* between pH 7 and 9, with slightly lower growth rates at pH 6, 10, and 11. The decline in growth rate at either extreme of the pH scale may be caused by changes in carbon availability, which would either interfere with photosynthesis or disrupt cell membrane processes [13].

Cyanobacterial activity tends to become limited under acidic conditions [39]. When the pH of a growth medium decreases beyond a certain threshold, the cellular pH of cyanobacteria also declines since cyanobacteria possess a limited ability to regulate their internal pH. Acidification of the cytoplasm thus ultimately reduces growth rate. Low growth rate may also be related to a low ATP/(ATP plus ADP) ratio under acidic conditions [40].

The reduction in *A. maxima* growth rate at a pH > 10 may be related to reduced CO_2_ availability in the culture medium. Reduced CO_2_ limits photosynthetic activity and increases the production of reactive oxygen species (ROS), which generate oxidative stress [13,37,41,42]. Cyanobacteria possess a CO_2_ concentrating mechanism that increases the cellular pool of inorganic carbon and the supply of CO_2_ to RuBisCo, thus promoting photosynthesis under alkaline conditions [41]. In the present study, however, the photosynthetic performance of *A. maxima* decreased at a pH > 9, indicating that a higher pH interferes with photosynthesis. In *A. platensis*, a highly alkaline pH induces the overproduction of antioxidants [13]. High concentrations of antioxidant enzymes and non-enzymatic molecules such as glutathione or vitamins can indicate oxidative stress [43]. Thus, the high values for these indicators in the study by Ismaiel, El-Ayouty, and Piercey-Normore [13] suggest that high pH conditions cause oxidative stress in *A. platensis*. Although we did not measure oxidative stress indicators in this study, oxidative stress represents a likely mechanism for the reduced growth of *A. maxima* observed at a higher pH.

Interestingly, *A. maxima* maintained slower but relatively stable growth (59.8% of maximal growth) at the lowest pH tested in this study (pH 4). In nature, cyanobacteria do not inhabit environments with a pH < 4 [36,44]. Low pH may damage cellular functions or increase the energy requirement of cells via a direct impact on cellular metabolism and membrane transport [45]. Abiotic stresses such as extreme pH also lead to the generation of ROS, resulting in severe consequences such as the deterioration of cellular metabolism and damage to cellular molecules such as proteins, DNA, and membrane lipids [13]. Algae have developed several mechanisms, such as the production of non-enzymatic and/or enzymatic antioxidants, to mitigate the effects of ROS. Enhancement of the antioxidant system in *Arthrospira* spp. has been reported previously, and cyanobacterial antioxidants have been investigated for industrial production [46]. The sustained growth of *A. maxima* cells under acidic conditions (pH 4) in this study suggests the existence of an unforeseen protective mechanism against highly acidic environments. If true, *A. maxima* would be a potential candidate for mass cultivation under the conditions of high CO_2_ and low pH caused by global warming.

Cultivation of cyanobacteria for wastewater treatment is gaining popularity since cyanobacteria can remove inorganic compounds such as nitrogen and phosphorous from agro-industrial runoff, thus removing pollutants while producing biomass [47,48]. However, cyanobacterial effectiveness depends on the pH of wastewater systems, with activity severely inhibited at pH extremes. Amendments may be used to maintain a suitable pH [49]; however, this process is expensive. The cultivation of *A. maxima* in wastewaters might offer a cost-effective solution since *A. maxima* showed a broad range of pH tolerance in this study, including a growth rate of 7.83%·d^−1^ at pH 4.

The specific growth rates of *A. maxima* cultures at different temperatures are presented in Figure 1B. *A. maxima* cultures showed the highest growth rate (circa 11.2–13.7%·d^−1^) at 30–35 °C and grew well (8.9%·d^−1^) at 25 °C. *A*. *maxima* cultures showed negligible growth (0.09%·d^−1^) after 8 d at 10 °C or 15 °C and died after 3 d at 5 °C. The optimum temperature for growth of *A. maxima* was consistent with that reported in a previous study [50]. *A. platensis* shows tolerance to a wide range of temperatures (20–40 °C), with maximum growth at 35 °C and no growth at 45 °C [51]. Ranjitha and Kaushik [52] obtained the maximum biomass of *Nostoc muscorum* at 30 °C and 35 °C. An analysis of different strains of *Arthrospira* revealed that the optimal temperature for growth varied from 35 °C to 37 °C, with growth severely restricted at 40 °C [53]. Temperature has a profound influence on metabolic processes, and the optimum temperature of a particular algal strain determines its productivity [54]. Cyanobacteria tend to have relatively greater temperature dependence and higher optimum growth temperatures than other phytoplanktons. This relatively high optimum growth temperature makes *A. maxima* a potential candidate species for future mass cultivation under an ocean-warming scenario.

### 3.2. Pigment Biosynthesis

After the exposure of *A. maxima* cultures to different pH conditions for 8 d, the concentration of Chl *a* (0.338–0.342 mg·g^−1^) was highest at pH 6–8 (Figure 2A). The lowest concentrations of Chl *a* were observed at pH 4 (0.181 mg·g^−1^) and pH 11 (0.103 mg·g^−1^). A previous study determined that the Chl *a* content of *A. platensis* is highest at pH 7 and lowest at pH 11 [55]. Among the pH levels tested for *Fischerella ambigua* (pH 5, 7, and 9), Chl *a* content was highest at pH 7 [56]. By contrast, the Chl *a* content of *Nostoc* spp. is not affected by pH [57]. In cyanobacteria, pH extremes change the charge status of proteins, causing their denaturation. Acidic conditions cause Chl *a* degradation [58]; however, no report has been published explaining reduced Chl *a* content under alkaline conditions. In spinach (*Spinacia oleracea*), a pH > 9 induces changes in the PSII membrane proteins related to the release of Cl^−1^ ions from the deprotonation of Cl^−1^-binding groups or its replacement with OH^−1^ ions [59,60]. As chlorophylls are embedded in the PSII membrane, structural changes in the PSII membrane proteins could explain the reduced Chl *a* concentration and Chl *a* fluorescence in *A. maxima* under alkaline conditions.

For phycobiliproteins, PC concentration was highest at pH 7 (1.98 mg·g^−1^), followed by pH 8 (1.96 mg·g^−1^), pH 6 (1.69 mg·g^−1^), pH 4 (0.452 mg·g^−1^), and pH 11 (0.336 mg·g^−1^) (Figure 2A). The concentration of APC was highest at pH 8 (1.142 mg·g^−1^), followed by pH 7 (0.974 mg·g^−1^), and lowest at pH 4 (0.148 mg·g^−1^) (Figure 2A). The concentration of PE was highest at pH 8 (0.561 mg·g^−1^), followed by pH 11 (0.120 mg·g^−1^), and lowest at pH 4 (0.102 mg·g^−1^) (Figure 2A). A previous analysis of the effect of pH on the phycobiliprotein content of 10 cyanobacterial strains belonging to *Anabaena* spp., *Calothrix* spp., *Nostoc* spp., and *Phormidium* spp. recorded maximal and minimal concentrations of phycobiliproteins at pH 6–7 and pH 9 (and pH 5, depending on the strain), respectively [12]. In *Nostoc* sp. UAM, alkaline conditions (pH 7.0–9.0) enhance the total phycobiliprotein content [57]. The pH required for the maximal production of phycobiliproteins in *A. maxima* was therefore consistent with the pH optima reported for other cyanobacterial species [2].

In *A. maxima*, pH 11 inhibited the production of phycobiliproteins and Chl *a*. Phycobiliproteins located in phycobilisomes comprise the major photosynthetic antenna of PSII, and any change in the membrane structure of PSII might also affect the content of phycobiliproteins. This implies that alkaline pH could have been responsible for the decrease in phycobiliproteins at pH 11 via the same mechanism as that underlying the change in Chl *a* concentration.

The effects of temperature on photosynthetic pigment content in *A. maxima* cultures are shown in Figure 2B. Both Chl *a* and PC concentrations were highest (0.428 and 2.445 mg·g^−1^, respectively) at 25 °C. The concentrations of these pigments did not vary significantly at 20 °C, 30 °C, and 35 °C. By contrast, we recorded the highest concentrations of APC and PE at 15 °C, and accumulation of these pigments showed no significant difference at 10 °C, 30 °C, and 35 °C (Figure 2B). No trace of cyanobacterial pigments was observed at 5 °C.

Temperature not only influences respiration rate, membrane stability, and nutrient uptake in cyanobacteria [61], but also cyanobacterial species’ diversity and metabolic product biosynthesis [62]. However, the effect of temperature on the production of chlorophyll and phycobiliproteins has not been extensively studied in cyanobacteria. In *A. platensis* grown at temperatures ranging from 20 °C to 40 °C, pigment contents increase with temperature up to 35 °C but then decline upon further increases in temperature [51]. Cultures of *A. platensis* display the highest Chl *a* fluorescence at 35 °C, which is 10 °C higher than the temperature required for maximal Chl *a* production.

The biosynthesis of phycobiliproteins in cyanobacteria is affected by temperature, with strain-specific temperature optima for production. In the present study, 25 °C was optimal for PC biosynthesis and 15 °C was optimal for APC and PE biosynthesis in *A. maxima* (Figure 2B). Temperatures between 30 °C and 37 °C are reported to be most suitable for phycobiliprotein production in most cyanobacteria, including some *Arthrospira* species [16,28,51,62,63]. The lowest temperature recorded for maximal phycobiliprotein production was in *Synechococcus* sp. PCC 7002, at 22 °C [64]. Thus, the response of phycobiliprotein production to temperature in *A. maxima* is more similar to that of *Synechococcus* than to that of *Arthrospira* spp., and *A. maxima* appears to possess the lowest optimum temperature reported for the production of phycobiliproteins [22]. The substantial production of PE and APC in *A. maxima* over a wide range of temperatures (10–35 °C) contrasts with a significant reduction in phycobiliproteins in *A. platensis* when grown at 20 °C compared with 35 °C [51]. This trait may make *A. maxima* more favorable than *A. platensis* for pigment production.

In general, cyanobacteria are highly adaptable to a wide range of temperatures and can be found growing in extreme environments. Nonetheless, an optimal temperature is important for biomass and pigment production in each species. Notably, no pigments were produced by *A. maxima* at 5 °C, which may be due to the effect of low temperature on some of its metabolic processes.

Phycobiliproteins may serve as a nitrogen reserve [65]. PC and PE act as nitrogen storage compounds in *A. platensis* and *Synechococcus* sp. strain DC2, respectively [66,67]. In the current study, the levels of PE and APC were higher at 15 °C than at 25–35 °C. However, we observed significant growth inhibition at 15 °C (only 4.3% of the maximum at 30 °C), concomitant with a 57.7% reduction in photosynthesis. The reduced photosynthetic capacity in *A. maxima* at 15 °C suggests that low temperature might cause the cyanobacterium to increase the accumulation of PE and APC for future use. This would enable cells to photosynthesize and grow immediately after environmental conditions become optimal, without requiring additional costs for the resynthesis of pigments. This opportunistic trait of *A. maxima* could be highly beneficial during outdoor cultivation under fluctuating environmental conditions.

### 3.3. Photosynthetic Activity

The chlorophyll fluorescence parameters (*F_v_*/*F_m_*, Φ_PSII_, and rETR_max_) for *A. maxima* under varying environmental pH are presented in Table 1. The values of all three parameters were highest at pH 7–8. However, *A. maxima* was still photosynthetically active, as determined by the chlorophyll parameters, even at pH extremes (pH 4 and pH 11).

The maximal values of *F_v_*/*F_m_* (0.43–0.44) were recorded at pH 7–8. Although the *F_v_*/*F_m_* values were reduced at a pH < 6 and pH > 9, these values were still within the range observed in healthy cyanobacteria. Other studies have reported *F_v_*/*F_m_* values ranging from 0.35 to 0.45 in *A. platensis* [68] and 0.3–0.4 in *Anabaena* spp. [69]. Decreases in the *F_v_*/*F_m_* ratio can indicate damage to the photosynthetic apparatus and a reduction in the photochemical efficiency of PSII [70]. In the present study, we observed a significant reduction in the *F_v_*/*F_m_* value at a pH > 8, indicating a decline in the photochemical efficiency of PSII.

The values of Φ_PSII_ in *A. maxima* were highest (0.35–0.38) at pH 7–8, and we observed a significant reduction in Φ_PSII_ under acidic (pH 4–5) and alkaline conditions (pH 11) (Table 1). The fluorescence parameter Φ_PSII_ measures the efficiency of PSII, representing the proportion of light used in photochemistry [70]. The decline in Φ_PSII_ observed under pH extremes suggests that PSII electron transport and the generation of NADPH and ATP were disrupted, inhibiting carbon fixation and assimilation via inhibition of the primary photochemical reactions. In this study, the decrease in Φ_PSII_ at pH 4–5 and pH 11 was also associated with a decrease in *F_v_*/*F_m_* and rETR_max_.

The values of rETR_max_ were greatest at pH 7 and 8, and significantly lower at a pH < 6 and a pH > 9 (Table 1). rETR_max_ is associated with photosynthetic electron transport activity. The decline in rETR_max_ at extreme pH levels suggests that *A. maxima* electron transport complexes were inhibited, and that NADPH and ATP production, as well as carbon assimilation, were constrained [71]. The reduced values of rETR_max_ in *A. maxima* at a pH < 6 and a pH > 9 may indicate that electron transport activity was substantially inhibited during photosynthesis, leading to a possible decline in PSII photochemical efficiency.

Among the various temperature treatments of *A. maxima*, the values of *F_v_*/*F_m_*, rETR_max_, and Φ_PSII_ were highest at 25–30 °C (Table 1). A comparison of means demonstrated that the values of *F_v_*/*F_m_* at temperatures ranging from 10 °C to 15 °C were significantly lower than those at 25–35 °C. Previous studies have reported that temperature stress inhibits the activity of PSII, leading to a significant reduction in *F_v_*/*F_m_* [72,73]. In this study, we observed a significant reduction in the *F_v_*/*F_m_* of *A. maxima* at 10–15 °C, indicating that lower temperatures result in the disruption of PSII reaction centers and impairment of photosynthetic activity. 

Φ_PSII_ was highest at 25 °C (0.27), significantly reduced at 10–15 °C (0.16–0.21), and zero at 5 °C (Table 1). A reduction in Φ_PSII_ values at temperatures lower than 15 °C indicates that photochemical utilization of light energy absorbed by *A. maxima* grown at these temperatures was lower than in *A. maxima* grown at 20–35 °C.

The values of rETR_max_ were significantly lower at temperatures ranging from 10 °C to 20 °C than at temperatures greater than 25 °C (Table 1). A decline in rETR_max_ values at 10–20 °C, as discussed above, implies that electron transport and carbon assimilation were inhibited [71].

### 3.4. Purification of PC

Many different techniques for the extraction and purification of phycobilins from cyanobacterial species have been reported (Appendix A). Figure 3 shows the protein extraction and purification process used in this study. As described in Materials and Methods, the purification process included fractionation, ultrafiltration, and ion exchange and size exclusion chromotography.

As illustrated by the electrophoresis results in Figure 4A, our proposed three step separation and purification process removed other proteins, leaving only the purified PC. We sent the crude PC extract and purified PC sample to an external testing center, which used the noise band detection method with HPLC and TOTALLAB software to confirm all proteins, including trace proteins not visually confirmed by electrophoresis. The results of the analysis of the crude PC extract and the purified sample after the three step separation procedure is shown in Figure 4B. Eleven peaks were observed in the HPLC profile of the crude extract, while only two peaks, at 8.6 min and 9.6 min, were observed in the sample that underwent three steps of extraction and purification. In addition, the peak at 8.6 min was found to correspond to 97.6% of the total protein in the purified sample, indicating that highly purified PC was obtained. The purity was confirmed to be 97.6%, and the final purified PC content was 1 mg·mL^−1^. To identify the cyanobacterial species with the highest phycocyanin yields, a reliable quantification method is needed. So far, phycobiliprotein yields or concentrations are most often calculated from absorbance values (Appendix A), and this relatively simple approach is sometimes misleading [74]. 

Therefore, a different approach is needed. In this study, we analyzed the protein bands from the extracts and determined the purity based on the relative band area of the total amount instead of reporting the A_pc_/A_280_ ratio. Appendix A lists the various examples of PC purification procedures described in the literature, including the protocol presented in this work, with the PC yield and purity obtained. The extraction yield of *A. maxima* in this study does not appear to be lower than that of other cyanobacterial species, including *A. platensis*.

## 4. Conclusions

*A. maxima* has several favorable traits for commercial production, including tolerance to a range of pH conditions and the hyper-accumulation of PE and APC at low temperatures. These traits may offer significant advantages for future exploitation, especially in outdoor cultivation with fluctuating pH and temperature. We believe that *A. maxima* has greater potential for the commercial production of biomass and phycobiliproteins than *A. platensis* because of its ability to adapt to wider ranges of pH and temperature.

The enhanced production of PE and APC at the cost of algal growth in *A. maxima* grown at low temperatures is similar to that observed in the same species exposed to low-intensity blue light [75]. These similar responses may be due to common regulatory mechanisms, or at least crosstalk in the expression of genes encoding phycobiliproteins, with the end result of optimizing the collection of light energy in anticipation of favorable environmental conditions (Figure 5). In higher plants, the blue light photoreceptor phototropin is known to perceive both blue light and temperature, and it uses this information to regulate photosynthesis [76]. The different behaviors of Chl *a* and PC vs. PE and APC in response to temperature should be investigated in future studies.

The PC market size was estimated at USD 155.3 million in 2020, and is predicted to reach USD 409.8 million by 2030, at a compound annual growth rate (CAGR) of 9.6% from 2021 to 2030 [77]. The price of PC may depend on its purity and intended use. Our study demonstrated a new method for the purification of PC from *A. maxima*, which produced PC at 1.0 mg·mL^−1^ with 97.6% purity. All processes represent economically viable alternatives for industrial scale-up if they can be adapted to a continuous process and simplified to allow for large-scale handling. 

The cost-effective production and harvesting of the biomass could be a major challenge for scaling up the purification of PC. To achieve large-scale production of the *A. maxima* biomass, new techniques such as photoautotrophic, mixotrophic, and heterotrophic production could be further explored [78,79].

There are a number of techniques for harvesting algae, such as sedimentation, flocculation, flotation, centrifugation, and filtration, or a combination of these. In general, the optimal harvesting method should use as few chemicals and as little energy as possible. Physical and chemisorption of algae with graphene (GO)-based tissue is also possible, since GO can easily interact with different functional groups [80,81,82].

**Figure 5 biology-11-00628-f005:**
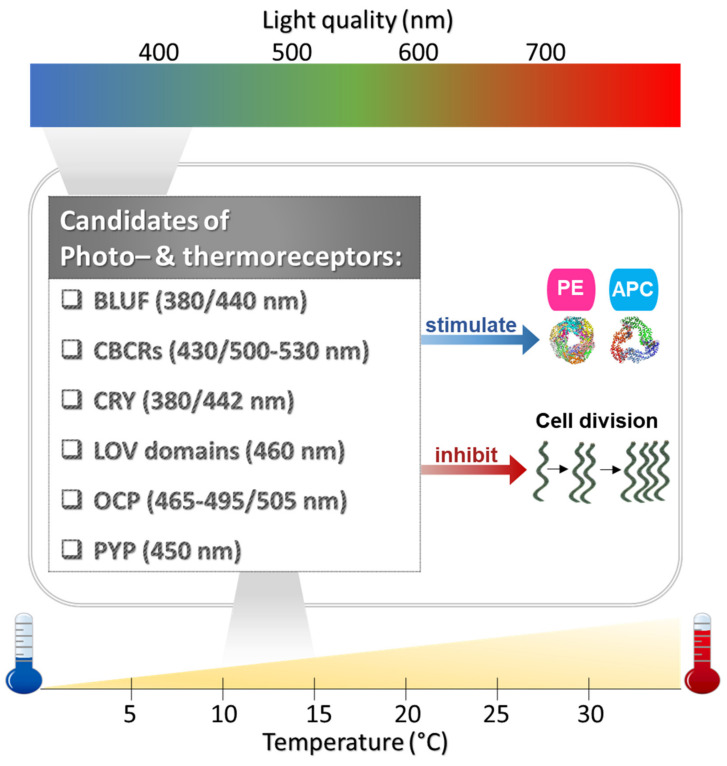
A conceptual framework for the possible simultaneous perception of blue light and low temperature by dual receptors in *A. maxima* that signal the production of phycoerythrin (PE) and allophycocyanin (APC) [83] and delay cell proliferation. BLUF: sensor of blue light using flavin adenine dinucleotide (FAD) [84], CBCRs: cyanobacteriochrome [85], CRY: cryptochrome [86], LOV domains: light–oxygen–voltage sensing domains [87], OCP: orange carotenoid-binding protein [88], PYP: photoactive yellow protein [89].

## Figures and Tables

**Figure 1 biology-11-00628-f001:**
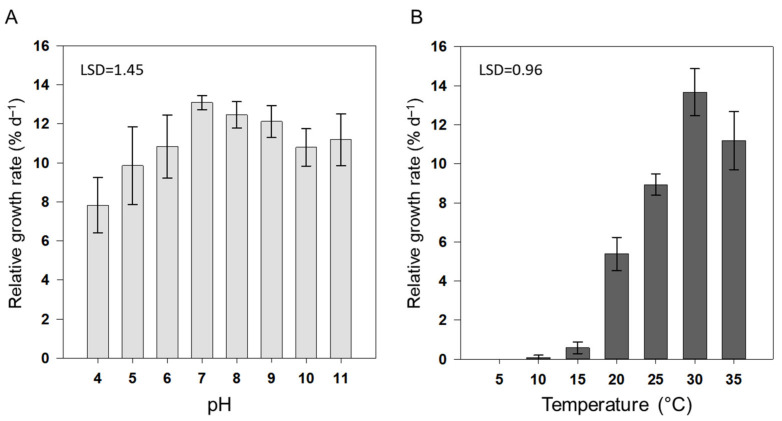
Effects of pH and temperature on the relative growth rate (RGR) of *Arthrospira maxima* cultures. Effects of various pH values (**A**) and temperatures (**B**) on RGRs. Data represent mean ± standard deviation (SD; *n* = 6).

**Figure 2 biology-11-00628-f002:**
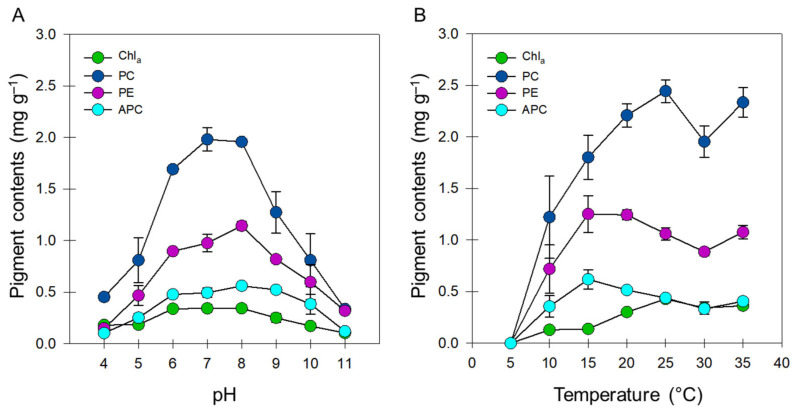
Effect of pH (**A**) and temperature (**B**) on pigment concentrations in *A. maxima* cultures. The concentrations of chlorophyll a (green) and phycobiliproteins including phycocyanin (navy), allophycocyanin (sky blue), and phycoerythrin (pink). Data represent mean ± SD (*n* = 6).

**Figure 3 biology-11-00628-f003:**
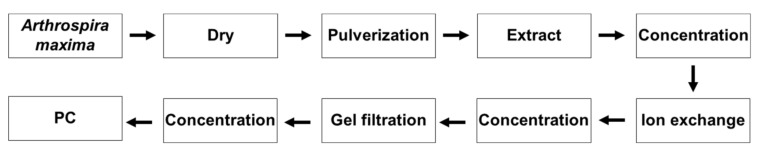
Procedure used to extract phycocyanin (PC) and obtain analytical grade PC.

**Figure 4 biology-11-00628-f004:**
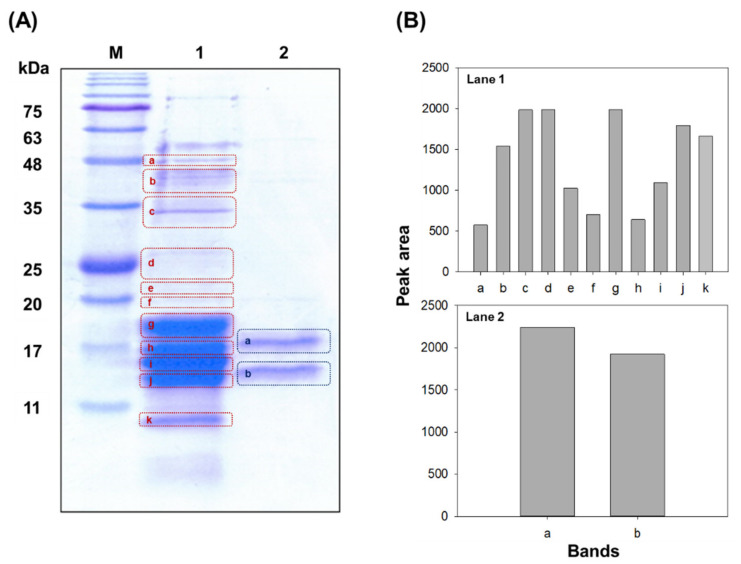
Purity of phycocyanin (PC) attained at crude extract and purification. SDS–PAGE of crude PC extract and purified PC from *Arthrospira maxima*. (**A**). Band peak areas of crude PC extract and purified PC (**B**). Lane M, protein marker (75, 63, 48, 35, 25, 20, 17, and 11 kDa); Lane 1, crude PC extract; Lane 2, purified PC.

**Table 1 biology-11-00628-t001:** Values of the fluorescence parameters in *Arthrospira maxima* cultures grown under different pH and temperature treatments.

Treatments	Fluorescence Parameters ^†^		
	*F_v_*/*F_m_*	rETR_max_	Φ_PSII_
pH 4	0.32 ± 0.019 ^a^	65.2 ± 8.54 ^a^	0.21 ± 0.008 ^ab^
pH 5	0.31 ± 0.020 ^a^	77.7 ± 9.40 ^a^	0.22 ± 0.006 ^abc^
pH 6	0.31 ± 0.016 ^a^	85.9 ± 9.53 ^a^	0.26 ± 0.017 ^c^
pH 7	0.43 ± 0.013 ^c^	118.2 ± 15.32 ^b^	0.35 ± 0.021 ^de^
pH 8	0.44 ± 0.011 ^c^	128.4 ± 16.64 ^b^	0.38 ± 0.012 ^e^
pH 9	0.35 ± 0.017 ^ab^	78.3 ± 9.43 ^a^	0.31 ± 0.021 ^d^
pH 10	0.37 ± 0.009 ^b^	74.4 ± 13.40 ^a^	0.25 ± 0.009 ^bc^
pH 11	0.32 ± 0.014 ^a^	58.54 ± 15.28 ^a^	0.18 ± 0.010 ^a^
5 °C	NA ^§^	NA ^§^	NA ^§^
10 °C	0.22 ± 0.021 ^ab^	40.9 ± 12.53 ^a^	0.16 ± 0.033 ^a^
15 °C	0.20 ± 0.020 ^a^	37.0 ± 13.89 ^a^	0.21 ± 0.019 ^b^
20 °C	0.30 ± 0.009 ^bc^	57.3 ± 9.28 ^a^	0.22 ± 0.034 ^bc^
25 °C	0.31 ± 0.039 ^c^	99.9 ± 6.79 ^b^	0.27 ± 0.035 ^d^
30 °C	0.33 ± 0.039 ^c^	87.4 ± 3.60 ^b^	0.25 ± 0.033 ^cd^
35 °C	0.31 ± 0.039 ^c^	88.75 ± 6.68 ^b^	0.21 ± 0.022 ^b^

^†^ Data represent mean ± standard deviation (SD; *n* = 6). Different superscript letters indicate significant differences (*p* < 0.05). *F_v_*/*F_m_*, maximum quantum yield; rETR_max_, maximum relative electron transport rate; Φ_PSII_, effective quantum yield; ^§^ NA, not applicable. Fluorescence parameters could not be measured at 5 °C because *A. maxima* cells died after 3 days of culture.

## Data Availability

All the results found are available in this manuscript.

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
