# Peer review of "Commercial Potential of the Cyanobacterium Arthrospira maxima: Physiological and Biochemical Traits and the Purification of Phycocyanin"

_biology, 2022, doi:10.3390/biology11050628_

Round 1

Reviewer 1 Report

This is a carefully planned and implemented comprehensive research on phycobilins from Arthrospira. I believe it would be of value to biotechnologists and practitioners.
Some issues to deal with:
Please give a reference or a full recipe for the SAG medium.
Please specify the algorithm the TOTALLAB software uses.
Section 2.4.3: what were the HPLC detection conditions?
L278: pH 6 is not an alkaline pH.
At which pH the data on Fig.1b were obtained? And at which temperature those on Fig. 1a were obtained? A similar question  can be raised regarding Fig. 2, Table 1.
Table 2 is very long; it can be painlessly put into the online supplementary material.
The part about the attempt of a tandem dye formulation can be omitted from the paper

Author Response

Thank you for your constructive comments, which have improved the clarity of our manuscript substantially.

Point 1: Please give a reference or a full recipe for the SAG medium.  

Response 1: We have added the information of the SAG medium. (line 140)

Point 2: Please specify the algorithm the TOTALLAB software uses.

Response 2: We have added the name of the developer of TOTALLAB software, TotalLab Ltd. (Newcastle upon Tyne, UK); however, the analysis was performed by KBSI (https://www.kbsi.re.kr/eng), as described in Section 2.4.3 Further separation by gel filtration (lines 329-334)

Point 3: Section 2.4.3: what were the HPLC detection conditions?

Response 3: HPLC detection conditions have been described in Section 2.4.3 as follows. An objective analysis of the purified proteins obtained from the three-step separation process was performed by the Korea Institute of Basic Science (KBSI, https://www.kbsi.re.kr/eng) using HPLC [Agilent 1200 Series with the Aeries WIDEPORE (3.6 µm) C4 column [150 ï‚´ 4.6 mm]). Solvents used were 0.1% trifluoroacetic acid in water and 0.075% trifluoroacetic acid in acetonitrile, and the flow rate was 1 mL·min‑1 with a column temperature of 40°C and were received from and certified by KBSI.

Point 4: L278: pH 6 is not an alkaline pH.

Response 4: Yes. pH 6 is not alkaline. Our intended meaning was that cyanobacteria are alkaliphilic and not that pH 6 is alkaline. We apologize for the lack of clarity.

Point 5: At which pH the data on Fig.1b were obtained?

Point 6: And at which temperature those on Fig. 1a were obtained?

Response 5 and 6: The test was carried out at pH 8.5 and 25°C, which are the pH and temperature for the stock culture of A. maxima, as described in 2.1. Cyanobacterium culture.

Point 7: A similar question can be raised regarding Fig. 2, Table 1.

Response 7: We have revised the Materials and Methods section as follows:

“Cultures were maintained at 25°C, pH 8.5, and a salinity of 20 psu. Cultures were exposed to 60 μmol m-2·s-1 photon flux density (PFD) light, with a 14/10 h light/dark cycle.” (lines 140-142)

“While testing the effect of a single environmental factor, all other conditions, including pH, temperature, salinity, PFD, and photoperiod, were kept constant, as described above. No time-series of pH monitoring was carried out.” (lines 152-154)

Point 8: Table 2 is very long; it can be painlessly put into the online supplementary material.

Response 8: Thank you. We have moved the Table 2 to the online supplementary material.

Point 9: The part about the attempt of a tandem dye formulation can be omitted from the paper.

Response 9: We have removed the following text related to the tandem dye formulation from the manuscript:

In Abstract section:

Attempts to develop a tandem organic dye using purified PC and the organic dye Cy7 was a failure, due to insufficient energy transfer from purified PC to Cy7, but it is worth developing tandem dyes using PC with other organic acceptor dyes to meet growing industrial demand.

In Introduction section:

‘Natural phycobiliproteins are excellent at absorbing and emitting light, and display low quenching energy loss and high quantum yield. These fluorochromes (or fluorescent dyes) are very useful for flow cytometry because they produce bright fluorescent signals with large Stokes shifts and can be easily coupled to antibodies [28]. Their strong fluorescence allows these fluorochromes to be used for measurements requiring high sensitivity, such as detection of cellular antigens expressed at low epitope density. Synthetic low molecular weight fluorochromes such as FITC (fluorescein isothiocyanate) and derivatives of cyanine (Cy) and rhodamine are widely used in flow cytometry. The separate development of fluorescent dye chemistry has promoted the general use of tandem dye conjugates, which emit bright fluorescence and have a large Stokes shift, allowing the simultaneous use of multiple fluorochromes with a single excitation source. Tandem dyes are prepared by covalently coupling a donor fluorochrome with various acceptor fluorochromes that participate in Foster (or fluorescence) resonance energy transfer (FRET). PE and APC are common FRET donor molecules, and their acceptor molecules are Texas Red, Cy5, Cy7, and their derivatives [28,29]. Fluorochromes derived from PE and APC, such as PE-Cy5, PE-Cy7, APC-Cy5.5, and APC-Cy7, are already commercially available.’

‘and then assessed the potential of the purified PC for use as a FRET donor molecule in combination with a Cy7 acceptor molecule in industrial flow cytometry.’

In 2.4.5 section:

‘2.4.5. Development of a tandem dye using purified PC and Cy7

To test the possibility of using the PC obtained from A. maxima for immunofluorescence or lab-on-a-chip, a tandem dye consisting of purified PC protein dissolved in sodium phosphate buffer (pH 7.0) and Cy7, an organic dye dissolved in DMSO, was prepared, aliquoted into tubes, and stirred at room temperature for at least 1 h. The stirred liquid sample was centrifuged through a filtration device with molecular mass cut-off 3,000 (Fisher Scientific) to obtain a supernatant after separation of low and high molecular mass proteins. After the supernatant was collected, it was placed in a fluorometer along with APC-cy7 (BD), and the fluorescence intensity emitted at 670-680 nm was measured after excitation at 635 nm.’

At the end of 3.4. Purification of PC section:

‘A dye produced by combining two dyes is called a "tandem dye," referring to a fluorescent substance that acquires new spectral properties through the FRET process. In the case of one tandem dye already on the market, the APC protein absorbs light energy at 635 nm and transfers it to Cy7 through the FRET process, producing a new emission at 785 nm. This is the spectral property of the tandem dye of APC and Cy7. However, in this study, we observed no fluorescence emission at 785 nm when a tandem dye made with PC separated and purified from A. maxima and Cy7 was excited with 635 nm light. The PC protein-specific fluorescence was high, but a relatively weak fluorescence peak was observed in Cy7, indicating that the light energy absorbed by the PC protein was not transferred to Cy7.’

At the end of conclusions:

‘It will also be worthwhile to further develop tandem dyes using PC with other organic acceptor dyes, to meet the growing industrial demand for flow cytometry, live cell staining, and multicolor immunofluorescent staining.’

Reviewer 2 Report

General comment

This study by Jihae Park et al. investigates the growth conditions for the production of target compounds and to ensure profitability in commercial applications of the cyanobacterium A. maxima. Additionally, the authors report a pipeline for the purification of phycocyanin from A. maxima.

The experiment is well designed, and the growth and phycobilin proteins yield of A. maxima are consistent with reported methodology and results.  Moreover, methodology for PC purification is appealing, thus I would like to suggest publication although at present the manuscript has still margin to be improved.

Specific comments

Simple summary

Line 31

C-PC : please explain the abbreviation and check accordingly throughout the text when mentioning PC (or C-PC) and APC.

Introduction

Line 56

The authors probably refer to A. platensis. Please correct the statement and provide new updated reference on the topic.

Line 57

Commercial powders obtained from A. platensis are not a “marine product”, due to freshwater habitat distribution of the species. Please clarify the sentence adding information about habitat and distribution for A. maxima.

Line 65
Most critical environmental factor for microalgal growth is light, that to date represents the major drawback for scale up and industrial employment of microalgae. I suggest to at least mention this essential environmental factor prior to discussing pH and temperature.

Line 83
In this context the authors can clarify the usage of “C-PC” mentioned above in “simple summary”.

Materials and methods

Materials and methods can be overall improved, i.e. giving more details about the experimental design and setup regarding A. maxima cultivation and treatments.

Line 124

2.1 I suggest to change subheadings to “A. maxima culture” or “Cultivation and maintainance of A. maxima”.

Line 125
Did the authors perform the experiments on more than one strain? Did they pre-screen A. maxima strains?
Please explain, adding opportune Culture Collection reference for the used strain(s).

Line 126
Please explain how the medium is modified or add opportune reference in the text.

Line 132
The description of experimental set up and cultures manipulation could be improved: my concerns regard:

1) please indicate the duration of each experiments, including further eventual manipulation i.e.

how many biological replicates? If six (line 265), how many used for destructive sampling for the analysis? Did the authors refresh medium over the 4 weeks? Did they monitor pH during the experiment?

2) I understand that due to the difficulty in counting A. maxima cells inoculum the authors assessed the relative growth rate within in a 8d experiment, while the treatments lasted for 4 weeks (line 209). Do the authors have any hint about the final yield of their cultures in terms of biomass and/or growth efficiency?

Line 255
the tandem dye is bound covalently?

Line 314
Specialized cyanobacteria can thrive also at a < 3 pH concentration; I also doubt that pH 4 could be defined as “extremely acidic”, especially due to the fact that the experiments range from pH 4 to 11, and the authors report a consistent growth rate at pH 4. Please modify accordingly.

Line 358
I suggest changing “In microalgae” into “In cyanobacteria”. However, several microalgae (es. the Rhodophytes Galdieria or Cyanidioschyzon) can thrive in thermoacidic environments at pH < 2.

Line 545
The authors describe hyper-accumulation of PE and APC, but it is not clear why they teamed the FRET dye with PC. I think it should be stated more explicitely within this context.

Line 562
in the conclusion section the use of C-PC abbreviation is misleading, please modify accordingly with respect of previous comments.

Minor comments

Line 69

Please add space after “[9].”

Line 81

Please add opportune reference in the text.

Line 312
Please use italics for species name and check throughout the text.

Author Response

Thank you for your constructive comments which have made our manuscript clearer and more rational.

Point 1: Simple summary  

Response 1: We have revised our summary as follows:

  1. maxima has been shown to be tolerant to a range of pH conditions and to exhibit hyperaccumulation of phycoerythrin and allophycocyanin at low temperatures. These properties could offer significant advantages for future use, especially in field cultivation with fluctuating pH and temperature. Our new method for purifying PC from A. maxima by ultrafiltration, ion exchange chromatography and gel filtration yielded PC in 1.0 mg-mL-1 with a purity of 97.6 %.

Point 2: Line 31 C-PC : please explain the abbreviation and check accordingly throughout the text when mentioning PC (or C-PC) and APC

Response 2: We have changed ‘C-PC’ to ‘phycocyanin (PC)’ and used the revised term throughout the manuscript. (lines 30,31,773,774)

Point 3: Line 56 The authors probably refer to A. platensis. Please correct the statement and provide new updated reference on the topic.

Response 3: Yes. Thank you for the precise comment. We have corrected line 24 ‘which has made it the second most important commercial photoautotroph after Chlorella’ to ‘which has made it one of the most important commercial photoautotrophs.

Point 4: Line 57 Commercial powders obtained from A. platensis are not a “marine product”, due to freshwater habitat distribution of the species. Please clarify the sentence adding information about habitat and distribution for A. maxima.

Response 4: ‘The growing international demand for high-value phytonutrients and pigments produced by A. maxima has made it the second most important commercial photoautotroph after Chlorella [6].’ has been changed to ‘The growing international demand for high-value phytonutrients and pigments produced by A. maxima has made it one of the most important commercial photoautotrophs together with Chlorella and A. platensis [6,7]. (lines 66-68)

Point 5: Line 65 Most critical environmental factor for microalgal growth is light, that to date represents the major drawback for scale up and industrial employment of microalgae. I suggest to at least mention this essential environmental factor prior to discussing pH and temperature.

Response 5: We have added new sentences on the importance of light as an environmental factor for cyanobacteria:

Among the various environmental factors that influence processes such as growth, development, reproduction, and photosynthesis in autotrophs, the quantity (irradiance) and quality (spectral composition) of light are essential. Light also affects the production of metabolites [10] (lines 76-80)

Point 6: Line 83 In this context the authors can clarify the usage of “C-PC” mentioned above in “simple summary”.

Response 6: As we have changed ‘C-PC’ to ‘PC’, it does not seem necessary to clarify the use of ‘C-PC’.

Point 7: Materials and methods can be overall improved, i.e. giving more details about the experimental design and setup regarding A. maxima cultivation and treatments.

Response 7: We have added more detailed information on the cultivation and treatments of A. maxima:

Square LED (Light Emitting Diode) panel lights (340 × 500 × 10 mm; Daewo, Bucheon, Korea) were used as main source of illumination for growth of A. maxima. Each LED strip comprised 20 diodes spaced at 1-cm intervals vertically and horizontally. PFDs were measured with a quantum sensor (Licor-1400, Li-Cor, Lincoln, NE, USA). (lines 142-146)

No time-series of pH monitoring was carried out. (line 154)

Point 8: Line 124  2.1 I suggest to change subheadings to “A. maxima culture” or “Cultivation and maintainance of A. maxima”.

Response 8: We have changed the subheading in line 137 to ‘A. maxima culture’.

Point 9: Line 125  Did the authors perform the experiments on more than one strain? Did they pre-screen A. maxima strains? Please explain, adding opportune Culture Collection reference for the used strain(s).

Response 9: Sorry for the confusion. We have used one strain provided from KMMCC and we have therefore added the strain number (LIMS-PS-1691). (line 138)

Point 10: Line 126 Please explain how the medium is modified or add opportune reference in the text.

Response 10: We have removed the word ‘modified’. (line 139)

Point 11: Line 132 The description of experimental set up and cultures manipulation could be improved: my concerns regard:

1) please indicate the duration of each experiments, including further eventual manipulation i.e. how many biological replicates? If six (line 265), how many used for destructive sampling for the analysis? Did the authors refresh medium over the 4 weeks? Did they monitor pH during the experiment?

2) I understand that due to the difficulty in counting A. maxima cells inoculum the authors assessed the relative growth rate within in a 8d experiment, while the treatments lasted for 4 weeks (line 209). Do the authors have any hint about the final yield of their cultures in terms of biomass and/or growth efficiency?

Response 11: The experimental period for testing the effects of pH and temperature is 8 days (line 157) and six replicates were carried out as in line 347, while for biomass production of A. maxima for purification of PC (line 300) it was 4 weeks. In both cases harvesting was done once, although in the latter case the medium was renewed at one week intervals. 200 mL of culture was harvested for the former experiments, while 8 mL of pelleted samples from centrifugation were used for the latter experiments.

We did not monitor pH after the experiments.

The reviewer is correct in saying that we assessed the relative growth rate within 8 days because of the difficulty in counting the number of A. maxima cells. In the case of the 4 week cultivation experiment to purify PC we have indicated that it was 613 mg as in line 301, which means 610 mg/800 mL (growth medium), i.e. 0.76 mg/mL. However, we did not measure the initial weight of the inoculum, and are not able to calculate the biomass efficiency.

Point 12: Line 255 the tandem dye is bound covalently?

Response 12: One of the reviewers asked us to delete the part of the manuscript on the tandem dye. Line 255 no longer exists. However, for the reviewer who asked a question about this, we would like to say yes and we understand that the tandem dye is known to be covalently bound.

Point 13: Line 314 Specialized cyanobacteria can thrive also at a < 3 pH concentration; I also doubt that pH 4 could be defined as “extremely acidic”, especially due to the fact that the experiments range from pH 4 to 11, and the authors report a consistent growth rate at pH 4. Please modify accordingly.

Response 13: We found two literatures stating that, in nature, cyanobacteria do not inhabit environments with pH < 4 [35,43] as in lines 427-429. On the other hand, we found that A. maxima maintained slower but relatively stable growth (59.8% of maximal growth) at the lowest pH tested in this study (pH 4). Therefore, we have simply emphasized that this alga can still grow well under acidic conditions at a pH of 4. However, on the advice of the reviewer, we deleted the word ‘extremely’ from the sentence ‘Sustained growth of A. maxima cells under extremely acidic conditions (pH 4) in this study’(line 438).

Point 14: Line 358 I suggest changing “In microalgae” into “In cyanobacteria”. However, several microalgae (es. the Rhodophytes Galdieria or Cyanidioschyzon) can thrive in thermoacidic environments at pH < 2.

Response 14: We have changed “in microalgae” to “in cyanobacteria” (line 472).

Point 15: Line 545 The authors describe hyper-accumulation of PE and APC, but it is not clear why they teamed the FRET dye with PC. I think it should be stated more explicitely within this context.

Response 15: For the same reason as above, we deleted the tandem dye part from the manuscript. In fact, we wanted to create a new type of tandem dye based on PC, which has hardly been researched so far.

Point 16: Line 562 in the conclusion section the use of C-PC abbreviation is misleading, please modify accordingly with respect of previous comments.

Response 16: Following the reviewer’s advice PC was used instead of C-PC throughout the manuscript.

Point 17: Line 69 Please add space after “[9].”

Response 17: We have inserted a space (line 85)

Point 18: Line 81 Please add opportune reference in the text.

Response 18: We have added a reference ‘Udayan et al. (2017)’ (line 81) (in line 96)

Point 19: Line 312 Please use italics for species name and check throughout the text.

Response 19: Thank you for this comment. We have corrected the species name to the italicized name. (in lines 68,644,789)

Reviewer 3 Report

Arthrospira maxima are one of the most potential algae for commercialization.  Exploring its optimal growth condition is an important step to accelerate its commercialization. This study investigated the effect of the range of pH and temperature on the growth and content of phycoerythrin and allophycocyanin. This study also explored the method to purify phycocyanin. The knowledge obtained from this study would help the biomass production in the outdoor facilities, and improve its commercial potential. It should attract attention from both academia and the industrial sectors. I would recommend its publication. A few comments and concerns are provided here:

  1. line 44: add “the” after “meet”.
  2. line 106-109: not quite understand what this sentence mean, please rewrite it!
  3. Do you measure the change of pH during the growth and at the time of sampling? I think it would be interesting to know how this algae to affect the environmental pH. The optimal pH(7-9) seems the initial pH. If we want to maintain an optimal growth, should we maintain this pH range? So, I recommend the authors include some results of the time course of their pH change during the experiment.
  4. what is the challenge for scaling up the purification of C-PC?

Author Response

Thank you for your constructive comments which have made our manuscript clearer and more rational.

Point 1: line 44: add “the” after “meet”.  

Response 1: We have already removed this sentence.

Point 2: line 106-109: not quite understand what this sentence mean, please rewrite it!

Response 2: Based on the reviewer’s comment to remove the part about tandem dye from the manuscript, we have removed this confusing sentence.

Point 3: Do you measure the change of pH during the growth and at the time of sampling? I think it would be interesting to know how this algae to affect the environmental pH. The optimal pH (7-9) seems the initial pH. If we want to maintain an optimal growth, should we maintain this pH range? So, I recommend the authors include some results of the time course of their pH change during the experiment.

Response 3: Thank you for the constructive comments on the time-series of measurements of pH change in the growth medium. We will certainly make this our next investigative step in the future.

Point 4: what is the challenge for scaling up the purification of C-PC?

Response 4: Thank you so much. After the reviewer’s comments we felt that it would be nicer to add the following sentences to improve the quality of our manuscript:

The cost-effective production and harvesting of biomass could be a major challenge for scaling up the purification of PC. To achieve large-scale production of A. maxima biomass, new techniques such as photoautotrophic, mixotrophic and heterotrophic production could be further explored [78,79].

There are a number of techniques for harvesting algae, such as sedimentation, flocculation, flotation, centrifugation and filtration, or a combination of these. In general, the optimal harvesting method should use as few chemicals and as little energy as possible. Physical and chemisorption of algae with the graphene (GO)-based tissue is also possible, since GO can easily interact with different functional groups [80-82].

Round 2

Reviewer 2 Report

The authors addressed all of my criticisms, therefore I support the publication of the article by Jihae Park and colleagues in its present form.